# A kinase-dependent feedforward loop affects CREBB stability and long term memory formation

Pei-Tseng Lee[1], Guang Lin[1], Wen-Wen Lin[1], Fengqiu Diao[2], Benjamin H White[2], Hugo J Bellen[1,3,4,5,6]*

[1]Department of Molecular and Human Genetics, Baylor College of Medicine, Houston, United States; [2]Laboratory of Molecular Biology, National Institute of Mental Health, National Institutes of Health, Bethesda, United States; [3]Howard Hughes Medical Institute, Baylor College of Medicine, Houston, United States; [4]Program in Developmental Biology, Baylor College of Medicine, Houston, United States; [5]Department of Neuroscience, Baylor College of Medicine, Houston, United States; [6]Jan and Dan Duncan Neurological Research Institute, Texas Children's Hospital, Houston, United States

**Abstract** In *Drosophila*, long-term memory (LTM) requires the cAMP-dependent transcription factor CREBB, expressed in the mushroom bodies (MB) and phosphorylated by PKA. To identify other kinases required for memory formation, we integrated Trojan exons encoding *T2A-GAL4* into genes encoding putative kinases and selected for genes expressed in MB. These lines were screened for learning/memory deficits using UAS-RNAi knockdown based on an olfactory aversive conditioning assay. We identified a novel, conserved kinase, *Meng-Po* (*MP, CG11221, SBK1* in human), the loss of which severely affects 3 hr memory and 24 hr LTM, but not learning. Remarkably, memory is lost upon removal of the MP protein in adult MB but restored upon its reintroduction. Overexpression of MP in MB significantly increases LTM in wild-type flies showing that *MP* is a limiting factor for LTM. We show that PKA phosphorylates MP and that both proteins synergize in a feedforward loop to control CREBB levels and LTM. key words: Drosophila, Mushroom bodies, SBK1, deGradFP, T2A-GAL4, MiMIC

DOI: https://doi.org/10.7554/eLife.33007.001

**\*For correspondence:**
hbellen@bcm.edu

## Introduction

Forward genetic screens coupled with classical conditioning paradigms (*Tully and Quinn, 1985*) have been successful in identifying many molecular determinants of learning and memory in *Drosophila* (*Guven-Ozkan and Davis, 2014*). Among the genes identified are several components of the cAMP signaling pathway, including *dunce* (a cAMP specific phosphdiesterase), *rutabaga* (a cAMP specific adenylyl cyclase), *Dc0* (Protein Kinase A, PKA) and *CREBB* (a transcription factor) (see *Figure 1—figure supplement 1* for these and others) (*McGuire et al., 2005*). In adult flies, these genes act in cells of the mushroom bodies (MB), higher brain centers responsible for associating conditioned stimuli (CS) and unconditioned stimuli (US), and for storing these associations (*Dubnau and Tully, 1998*). Associations are stored by the MB Kenyon cells (KC), which are activated by CS—such as odors—via cholinergic transmission from olfactory projection neurons and US—such as electric shock—via dopamine signaling (*McGuire et al., 2005*). Acetylcholine receptors permit $Ca^{2+}$ entry into KCs, while dopamine receptors activate a $Ca^{2+}$/Calmodulin-responsive adenylyl cyclase (*rutabaga*), which acts as a coincidence detector (*Busto et al., 2010*). cAMP produced by Rutabaga activates PKA and triggers a downstream MAP kinase cascade, which leads to the phosphorylation and

activation of the transcription factor CREBB (*Impey et al., 1999*). CREBB binds to cAMP response elements (CRE), activating the transcription of genes required for long term memory formation (*DeZazzo and Tully, 1995*).

CREBB is phosphorylated by several kinases, including PKA and CamKII, two kinases known to be involved in memory formation (*Horiuchi et al., 2004*; *Mayr and Montminy, 2001*) (*Figure 1—figure supplement 1*). Here we describe the identification of a novel gene *Meng-Po* (*MP*), that is required for memory formation but not for learning. The MP protein regulates the stability of CREBB together with PKA. In the absence of MP the protein stability of CREBB is affected, and removing a single copy of *MP* and *PKA* leads to a dramatic loss of CREBB and memory formation. In addition, overexpression of MP strongly promotes memory formation, indicating that MP is not only required but that it also plays an instructive role in memory formation.

## Results

### Genes encoding protein kinases expressed in MB in adult brain and behavioral consequences of RNA interference

To identify novel protein kinases involved in learning/memory, we developed a strategy that allows us to determine which kinases are expressed in MBs. We selected 27 putative kinase-encoding genes for which fly lines were available that contained intronic insertions of the M̲i̲nos M̲ediated Integration C̲asette (MiMIC) (*Nagarkar-Jaiswal et al., 2015*; *Venken et al., 2011*). These MiMICs contain a swappable cassette, allowing integration of any DNA using r̲ecombinase-m̲ediated c̲assette e̲xchange (RMCE). We replaced these cassettes with a Trojan exon encoding *SA-T2A-GAL4* to permit the detection of cells expressing the kinase-encoding genes using *UAS-mCD8::GFP* (*Diao et al., 2015*). Of the 27 putative protein kinase genes screened, we found 12 that are expressed in MBs (*Figure 1a*; *Figure 1—figure supplement 2*). To determine if these 12 kinases expressed in MB play a role in learning and memory we knocked down their expression using *UAS-RNAi* expressed under the control of the MB-selective *OK107-GAL4* driver (*Figure 1—figure supplement 3a*) and tested the performance of learning and 3 hr memory in knockdown flies upon an olfactory aversive conditioning assay (*Figure 1*; *Figure 1—figure supplement 3b*). In *Drosophila*, Pavlovian olfactory aversive learning requires coincidence detection of a conditioned stimulus (CS), an odor, and an unconditioned stimulus (US), an electric shock. Through a single training session using a T-maze assay in which flies are exposed to 12 CS-US pairings in one min, flies can associate CS with US and learn to avoid the odor paired with electric shock (*Tully and Quinn, 1985*). After training, flies can learn (tested immediately) and form an intermediate-term/3 hr memory (tested after 3 hr) (*Margulies et al., 2005*). We identified two putative kinases, *CG11221* and *wallenda* (*wnd*), which when knocked down, cause significant reductions in performance index for 3 hr memory, but not for learning (*Figure 1b–c*). Overexpression of *wnd* has been documented to enhance memory in *Drosophila* (*Huang et al., 2012*), but *CG11221* has not been previously characterized in flies. We therefore tested the effects of knocking down *CG11221* expression using a second RNAi expressed specifically in MB neurons to confirm the memory loss phenotype (*Figure 1—figure supplement 3b*). To provide additional evidence that *CG11221* is not required for learning, we used a short CS/US association protocol (30' CS + US instead of 60') to train flies. Knockdown of MP did not affect learning, further indicating that *MP* is not required for learning (*Figure 1—figure supplement 3b*).

*CG11221* is an evolutionarily conserved serine/threonine protein kinase (human *SBK1*: 37% identity and 75% similarity, *Figure 1—figure supplement 4*). *SBK1* is expressed in the hippocampus, cortex, and cerebellum of adult rodents and loss of *SBK1* in mice causes partial embryonic lethality (*Nara et al., 2001*; *Skarnes et al., 2011*). The *CG11221^{MI03008}-T2A*-GAL4; UAS-mCherry flies exhibit broad expression of the gene in third instar larvae and adult flies (*Figure 1—figure supplement 5*). As shown in *Figure 1a*, the gene is also expressed in the adult brain and is prominent in the MB. In light of its importance in memory, we renamed *CG11221, Meng-Po* (*MP*), for the Lady of Forgetfulness, a character in Chinese mythology who ensures that people are ready for reincarnation by providing the 'Tea of Forgetfulness' so they lose the memory associated with their former life.

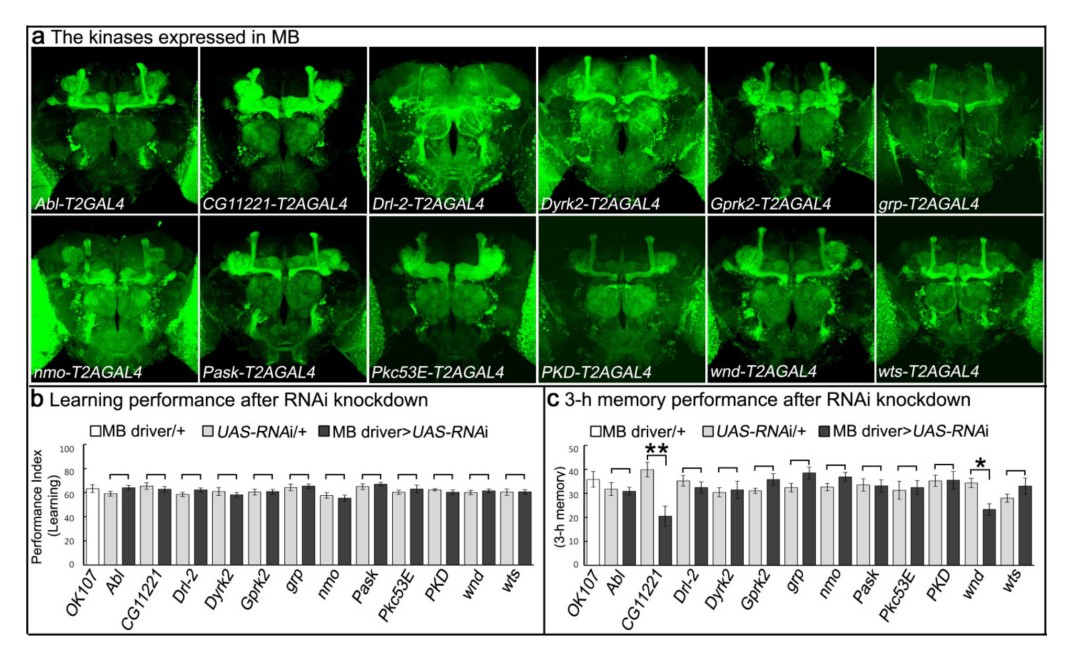

**Figure 1.** Genes encoding kinases expressed in MB of adult brains and behavioral consequences of RNA interference. (a) Expression patterns of genes encoding protein kinases in adult brains. The 12 protein kinases shown here are expressed in MB. (b–c) Screening results of behavioral assays by RNAi knockdown. The *MB-GAL4* driver, *OK107* was used to drive the *UAS-RNAi*. Flies were raised at 18°C until eclosion, transferred to 25°C for 3 days and behavioral assays were performed to test (b) learning and (c) 3 hr memory. The mean ±SEM is plotted for each genotype; n = 8 for each group. *p<0.05, **p<0.01.

DOI: https://doi.org/10.7554/eLife.33007.002

The following figure supplements are available for figure 1:

**Figure supplement 1.** Overview of the proteins required for olfactory aversive learning/memory formation in *Drosophila*.
DOI: https://doi.org/10.7554/eLife.33007.003

**Figure supplement 2.** The *T2A-GAL4* expression patterns of protein kinases that are not expressed in MB.
DOI: https://doi.org/10.7554/eLife.33007.004

**Figure supplement 3.** Reducing the levels of *CG11221* in MB affects 3 hr memory formation.
DOI: https://doi.org/10.7554/eLife.33007.005

**Figure supplement 4.** Comparison of protein sequence between human SBK1 and fly MP (CG11221).
DOI: https://doi.org/10.7554/eLife.33007.006

**Figure supplement 5.** *MP* is expressed ubiquitously.
DOI: https://doi.org/10.7554/eLife.33007.007

## Loss of MP causes a loss of memory

To determine whether the memory deficits resulting from loss of *MP* are due to its activity in adult MB neurons or are a result of its developmental expression, we used the deGradFP method (*Caussinus et al., 2011*; *Nagarkar-Jaiswal et al., 2015*) to selectively decrease levels of MP protein in MB of 5 day old flies. To do so, we replaced the MiMIC insertion of the *CG11221^03008* in the first coding intron with a *SA-GFP-SD* in-frame with the *MP* coding sequence using a previously described technique (*Nagarkar-Jaiswal et al., 2015*). This manipulation insures the expression of an internally-tagged MP-GFP-MP (MP-GFP) fusion protein. Flies homozygous for the modified *MP-GFP* gene were viable and anti-GFP staining of homozygous *MP-GFP* flies revealed expression in MB (*Figure 2a*). Furthermore, these flies did not exhibit obvious viability or learning/memory defects (*Figure 2—figure supplement 1*), indicating that the MP-GFP fusion protein retains its function. Crossing *MP-GFP* flies to flies that express *UAS-deGradFP* under the control of the MB-specific GAL4 driver *P247* (*Figure 2—figure supplement 2*) resulted in progeny in which the MP-GFP fusion protein was substantially depleted from MB when flies were raised at 28°C, but not at 18°C. By shifting flies between these temperatures we can reversibly eliminate the tagged protein from MB

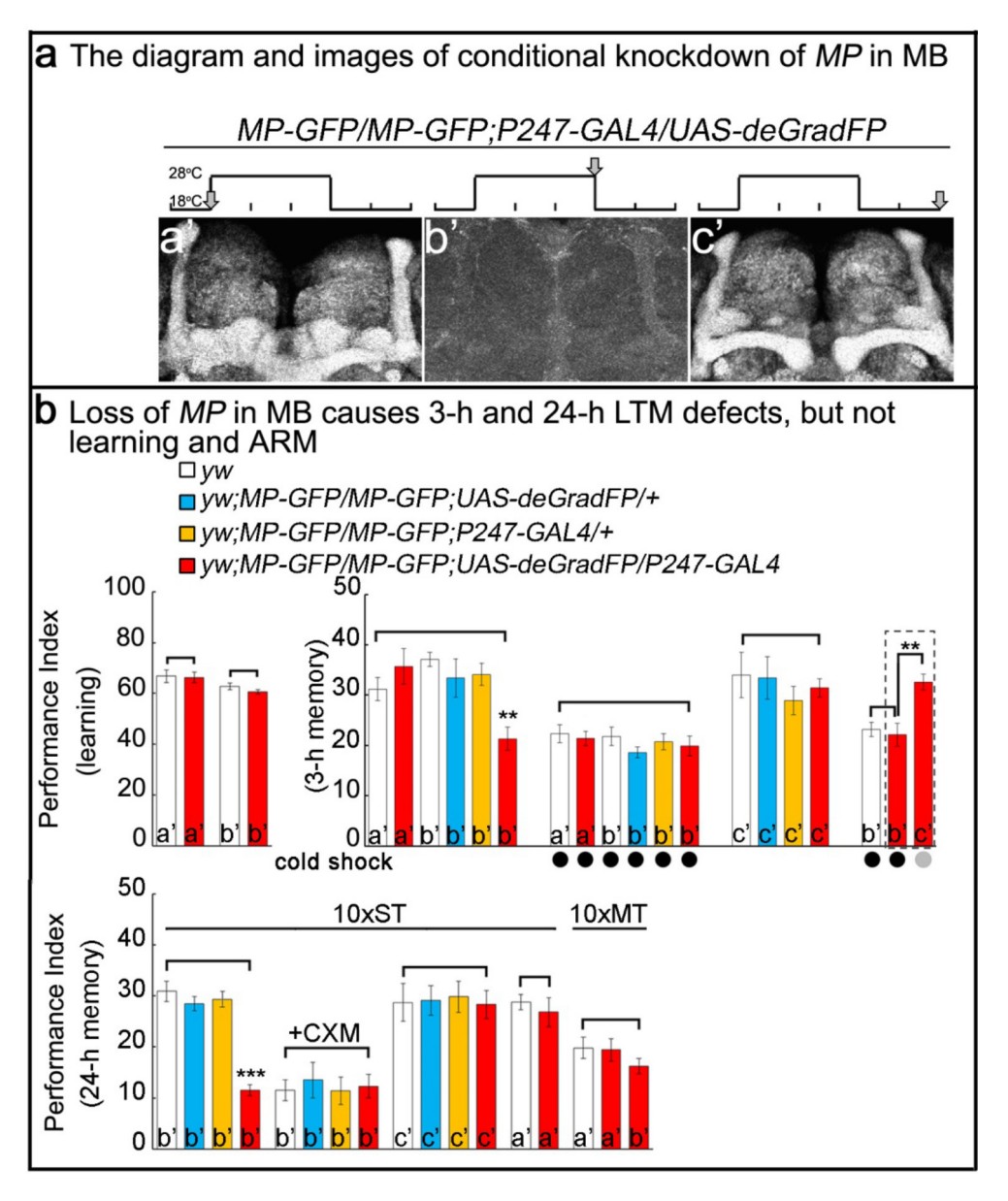

**Figure 2.** Loss of MP causes a loss of memory. (a) *UAS-deGradFP* driven by MB driver, *P247-GAL4*. a'-c': scheme for temporal control of deGradFP expression via temperature shift. (a') 18°C; (b') 28°C for three days; (c') 28°C for three days, followed by a shift to 18°C for two days. MP-GFP-MP expression at the time points shown above by arrows. (b) Performance scores of learning, 3 hr memory and 24 hr memory. (a'), (b') and (c') are referred as the time points in (a). (b), learning) Learning is normal after knockdown of MP in MB by deGradFP at 28°C for 3 days (b'). Flies raised at 18°C are used as a control (a'). (b, 3 hr memory) 3 hr memory is impaired after knockdown of MP in MB (b'). However, the performance score of ARM is intact in MP knockdown flies which are treated with a cold shock and compared to flies raised at 18°C (3 hr memory a' and b'+cold shock). In c' condition, the flies show normal performance score of 3 hr memory. The 3 hr memory impairment (in b'+cold shock, right panel) can be fully rescued when the animals are shifted to 18°C for two days (the groups boxed in dashed line are the exactly same flies). (c), 24 hr memory) 24 hr LTM (10 x ST) is impaired upon knockdown of MP in MB (b', red). After treatment with 35 mM cycloheximide (CXM), the MP knockdown flies (b'+CXM, red) don't exhibit a performance that is worse than control flies. In the c' conditions, the flies exhibit a normal performance score for the 24 hr memory assay. The performance of ARM (10 x MT) is intact. 10 x ST:10 times spaced training. 10 x MT:10 times massed training. The mean ±SEM is plotted for each genotype; n = 8 for each group. **p<0.01. ***p<0.001.
DOI: https://doi.org/10.7554/eLife.33007.008

*Figure 2 continued on next page*

*Figure 2 continued*

The following figure supplements are available for figure 2:

**Figure supplement 1.** *MP-GFP-MP* animals have proper learning and memory.
DOI: https://doi.org/10.7554/eLife.33007.009
**Figure supplement 2.** *P247-GAL4* was used for protein knockdown.
DOI: https://doi.org/10.7554/eLife.33007.010

neurons. Hence, the protein is present during development and early adulthood (*Figure 2a,a'*), avoiding developmental requirements. The temperature shift leads to a loss of MP in 5 day old flies (*Figure 2a,b'*). Upon a two day recovery period at 18°C, the protein levels are restored (*Figure 2a, c'*). In summary, the deGradFP technique with the *MP-GFP* flies provides a precise tool for manipulating and monitoring MP levels in specific tissues in adult flies in a reversible manner.

Using this strategy, we reduced the level of MP-GFP in adult MB for three days (*Figure 2a,b'*) and assayed learning and memory performance in these flies. As shown in *Figure 2b*, flies with MP depleted in the MB (b', red column) learn as well as *y w* control animals (b', white). However, when tested 3 hr after training, these flies exhibited severe memory deficits (*Figure 2b*, 3-h memory, middle panel, b', red column) when compared with either *y w*, other controls or animals of the same genotype tested prior to depletion of MP (*Figure 2b*, 3-h memory, middle panel). These results are consistent with those obtained by constitutive knockdown of *MP* expression using RNAi and demonstrate that MP is required in MB neurons specifically in adults (*Figure 1c*; *Figure 1—figure supplement 3b*).

It has been previously shown that the 3 hr memory has two distinct components: an anesthesia-resistant memory (ARM) and an anesthesia-sensitive memory (ASM), only the latter of which can be erased by cold-shock (*Lee et al., 2011*; *McGuire et al., 2005*). To determine whether MP functions in ARM, ASM or in both, we subjected trained animals in which MP-GFP had been depleted in the MB to cold-shock and found that the residual memory was unaffected (*Figure 2b*, 3-h memory, b' with cold shock). Furthermore, the memory performance in these animals was statistically indistinguishable from that of control animals subjected to cold shock. These results suggest that MP is required for ASM, but not for ARM. Importantly, loss of 3 hr memory can be restored by placing the flies at 18°C for two days after knockdown (*Figure 2b*, 3-h memory, c'). To test whether the transient loss of MP in the MB has long-term consequences to the animal's ability to form memories, we restored expression of MP-GFP after 3 days of depletion at 28°C by returning animals to 18°C. Their ability to learn and remember upon restoration of MP-GFP revealed complete recovery of memory function (*Figure 2b*, 3-h memory, right panel, column c').

In parallel studies we also tested animals with reduced MP for deficits in 24 hr long-term memory, LTM, and *radish* dependent 24 hr ARM (*Margulies et al., 2005*). LTM requires *CREBB* as well as new protein synthesis (*Fropf et al., 2013*; *Keene and Waddell, 2007*) and is typically induced by repetitive (10x) spaced training (ST) at 15 min intervals. In contrast, 24 hr ARM does not require protein synthesis (*Dubnau and Tully, 1998*; *Keene and Waddell, 2007*) and is induced by repetitive mass training (MT) without the 15 min resting intervals. We found that MP is dispensable for 24 hr ARM (*Figure 2b*, 24-h memory, 10x MT), but is required for LTM (*Figure 2b*, 24-h memory, 10x ST, b', red column). After treatment with 35 mM cycloheximide (CXM), an inhibitor for protein synthesis, the MP knockdown flies (*Figure 2b*, 24-h memory, 10x ST, b'+CXM, red column) do not perform worse than other control flies. This indicates that the loss of memory is protein synthesis-dependent. Loss of 24 hr LTM memory can be restored when returning the flies to 18°C for two days after knockdown. (*Figure 2b*, 24-h memory, 10x ST, c'). Hence, loss of MP affects neither learning nor ARM, but severely impairs 3 hr ASM and 24 hr LTM.

## Overexpressing MP enhances 24 hr LTM and increases CREBB activity

To determine whether MP is not only necessary for LTM, but also acts as a limiting factor for LTM, we overexpressed MP in MB and analyzed 24 hr LTM. To avoid saturating LTM, as occurs when the spaced training is repeated 10x, we adopted a paradigm in which flies were subjected to only a 3X training paradigm (*Lee et al., 2011*) (3x ST). Conditional overexpression of MP in MB under the control of *OK107-GAL4* was accomplished using a temperature-sensitive GAL4 inhibitor (*Tub-GAL80^{ts}*),

which blocks *UAS-MP* expression at 18°C (*Figure 3a,a'–b'*), but not 29°C (*McGuire et al., 2004*). Flies grown at 18°C and placed at 29°C for two days robustly expressed MP as detected by Western blot (*Figure 3a,c'*).

Flies were subjected to temperature protocols that either did (*Figure 3b,b'*) or did not (*Figure 3b,a'*) induce *MP* overexpression and we then tested their performance in both learning and LTM using the T-maze assay. The learning scores of *Tub-GAL80ts/+;UAS-MP-HA/+;OK107-GAL4/+* flies temperature-shifted to 29°C, and therefore overexpressing MP, were the same as those of unshifted flies of the same genotype and those of other control groups (*Figure 3b,c'*). In contrast,

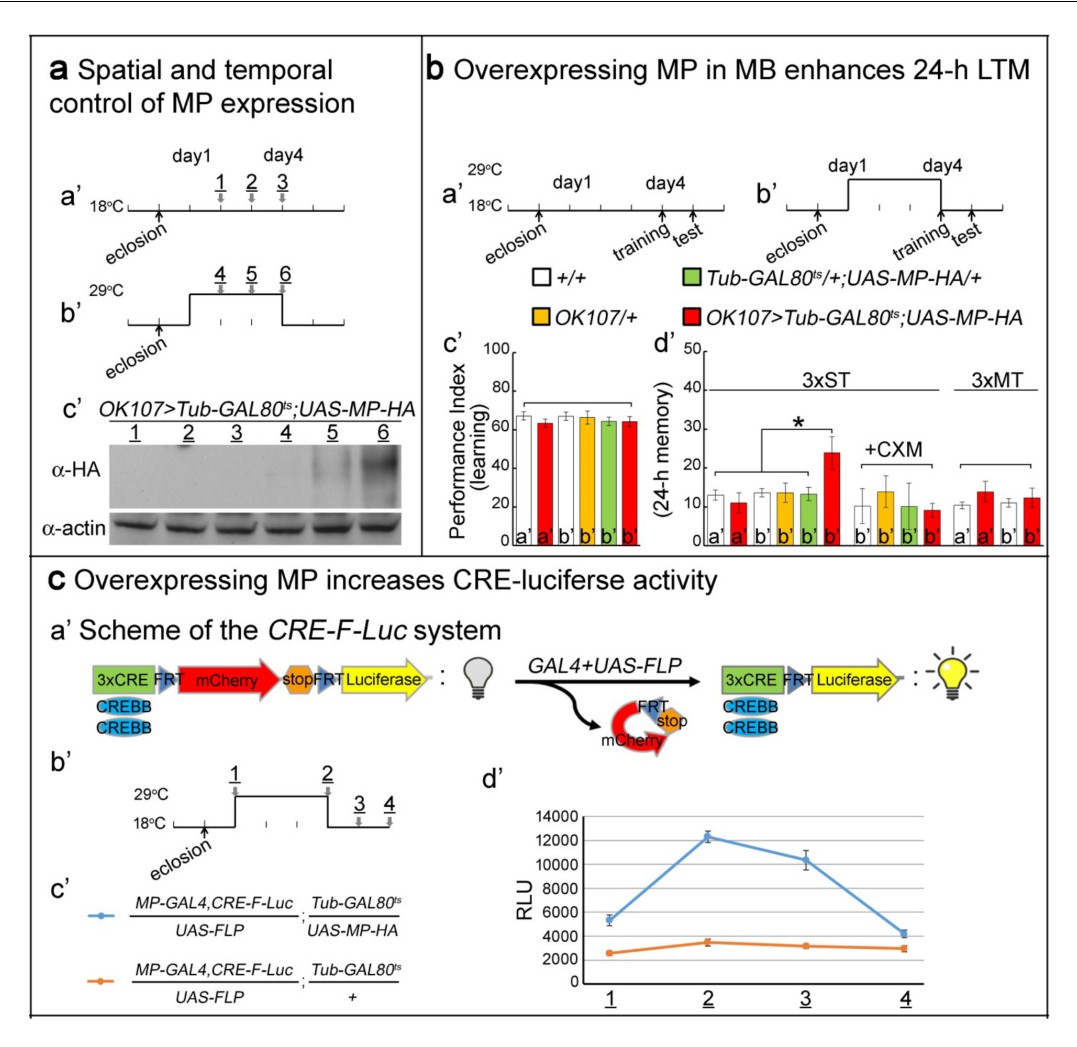

**Figure 3.** Overexpressing MP enhances 24 hr LTM and increases CREBB activity. (a) Spatial and temporal control of MP expression was achieved with a *MB-GAL4* driver and *Tub-GAL80ts*. Schemes are shown in (a') and (b'). MP expression was assessed by Western blot with α-HA antibody in (c'). (b) Temporal overexpression of MP in MB enhances 24 hr LTM. Diagrams for MP overexpression, training and testing of behavior are shown in (a') and (b'). The learning scores are normal after MP overexpression in (c') (red column, (b'). 24 hr LTM is significantly enhanced after 3x ST training in (d') (red column, (b'). However, memory enhancement can be erased by feeding flies with cycloheximide. The 24 hr ARM is unaltered after MP overexpression in (d') (red column, (b'), a'=no overexpression, b'=overexpression, and 3x ST (spaced training) or 3x MT (massed training). The mean ±SEM is plotted for each genotype; n = 8–12 for each group with 200 animals per group. *p<0.05. (c) Overexpressing MP increases CRE-luciferase activity. The *CRE-F-Luc* system is shown in (a'). The diagram is shown in (b'). The fly genotypes are shown in (c'). blue curve: *MP-GAL4,CRE-F-Luc/UAS-FLP;Tub-GAL80ts/UAS-MP-HA*. orange curve: *MP-GAL4,CRE-F-Luc/UAS-FLP;Tub-GAL80ts/+*. In (d'), the luciferase activity of the two genotypes of flies at the following time points: 1, 2, 3 and 4 which are shown in (b'). (1) at 18°C for one day after eclosion. (2) at 29°C for three days. (3) at 29°C for three days, then shift to 18°C for one day. (4) at 29°C for three days then shift to 18°C for two days. 10 fly heads (five males/five females) were collected for a single assay. RLU: relative luminescence unit. The mean ±SEM is plotted for each point; n = 6.
DOI: https://doi.org/10.7554/eLife.33007.011

the performance scores of these flies for 24 hr LTM were significantly enhanced after MP overexpression in MB (3x ST; *Figure 3b,d'*). The observed memory enhancement can be erased by feeding flies 35 mM CXM, indicating that enhanced memory is protein synthesis-dependent (3x ST +CXM; *Figure 3b,d'*). Moreover, this enhancement is specific to 24 hr LTM, and not 24 hr ARM induced by 3-times mass training (3x MT; *Figure 3b,d'*). Thus, MP is not only necessary for LTM, but also is also promoting LTM formation.

Given that *CREBB* is a central player in LTM formation, we tested its activity in MB during *MP* overexpression using a previously described luciferase assay. This assay relies on CREBB-mediated transcription of luciferase from the *CRE-F-Luc* construct, which contains three copies of the CREBB-binding cAMP Response Element (CRE), upstream of an FRT-flanked mCherry-encoding stop cassette and the firefly luciferase gene (*Figure 3c,a'*) (*Tanenhaus et al., 2012*). Spatial control of luciferase expression can be achieved by using the *GAL4/UAS* system to drive the cell-type specific expression of a *UAS-FLP* recombinase (*Figure 3c,a'–b'*). We created flies of the following genotype: *MP-T2A-GAL4,CRE-F-Luc/UAS-FLP;Tub-GAL80^{ts}/UAS-MP-HA* (*Figure 3c,c'*). In these flies, *GAL4* drives expression of *UAS-MP-HA* and *UAS-FLP* when the flies are shifted to 29°C. FLP removes the stop cassette in the *CRE-F-Luc* construct, allowing translation of luciferase in the *MP*-expressing neurons (*Figure 3c,a'–b'*). When assayed for luciferase activity, flies placed at 29°C for 3 days display a very significant increase—measured in relative luminescence units (RLU)—when compared to the flies kept at 18°C. The RLU gradually decrease to base line upon return to 18°C (*Figure 3c,d'*, blue). In contrast, the luciferase activity of control flies that do not carry the *UAS-MP* (genotype: *MP-T2A-GAL4,CRE-F-Luc/UAS-FLP;Tub-GAL80^{ts}/+*) display no change in RLU when subjected to the same temperature shifts (*Figure 3c,c'–d'*, orange). In summary, overexpression of MP in MB significantly enhances 24 hr LTM and increases CRE-luciferase activity. Our results are consistent with a model in which *MP* facilitates LTM by positively regulating CREBB activity (*Dubnau and Tully, 1998*).

## MP is a kinase, and loss of MP affects CREBB protein levels in MP gene trap animals

Given that *MP* encodes a putative kinase, we tested its ability to phosphorylate a known substrate of two other kinases, ERK and PKA. We expressed MP-HA in S2 cells, affinity purified the protein, and performed kinase assays on Myelin Basic Protein (MBP) (*Martenson et al., 1983*). As shown in *Figure 4a*, ERK, PKA and MP phosphorylate MBP. Given the CRE-luciferase assay results and the established regulation of CREBB by phosphorylation (*Horiuchi et al., 2004*; *Tully et al., 2003*), we hypothesized that CREBB is a substrate of MP. To determine if MP can phosphorylate CREBB we purified the CREBB protein from S2 cells and performed kinase assays. Unlike PKA, which is able to phosphorylate CREBB, MP was not able to phosphorylate CREBB (*Figure 4b*). Hence, although CREBB activity is dependent on MP levels, CREBB may not be a direct substrate of MP.

To more generally assess how *MP* upregulates *CREBB*, we sought to determine how CREBB activity is affected by loss of *MP* function. The *MI03008* MiMIC insertion in the *MP* gene functions as a gene trap (*Figure 4—figure supplement 1a*) and RT-PCR fails to amplify a product between these exons (*Figure 4—figure supplement 1b*). This and other evidence suggest that *MI03008* is a severe loss of function allele of *MP*. To estimate the levels of CREBB in *MI03008* mutants, we probed Western blots of protein isolated from the heads of adult *MI03008/MI03008* animals using an anti-phospho-CREBB antibody (*Fropf et al., 2013*) and compared the signal to that observed in Western blots from heads of control animals. In parallel, we analyzed the signals obtained using a second antibody (Pan-CREBB) that assesses total CREBB protein levels. Both antibodies revealed an approximately 60% reduction in immunoreactivity in samples from *MP* mutant brains (*Figure 4c,a'–b'*). Note, that there is a minor reduction in the ratio of phospho-CREBB/total CREBB (*Figure 4c,c'*). In contrast, CREBA protein levels are not altered (*Figure 4d*), consistent with the observation that *CREBA* has no role in memory formation (*Abrams and Andrew, 2005*). Finally, *CREBB* mRNA levels are unchanged in *MP* mutants (*Figure 4—figure supplement 1c*), indicating that the observed reduction in CREBB protein is a post-transcriptional effect.

We also tested if ATG2-CREBB is affected. This CREBB isoform is encoded by a downstream, in-frame initiation codon of one of the transcripts of the *CREBB* gene, and corresponds to a transcriptional activator. It has been shown to be required for memory enhancement (*Tubon et al., 2013*). As shown in *Figure 4—figure supplement 1d*, we find that both the anti-Pan-CREBB antibody and the antibody recognizing ATG2-CREBB (*Fropf et al., 2013*) identify a ~30 kDa band that is reduced in

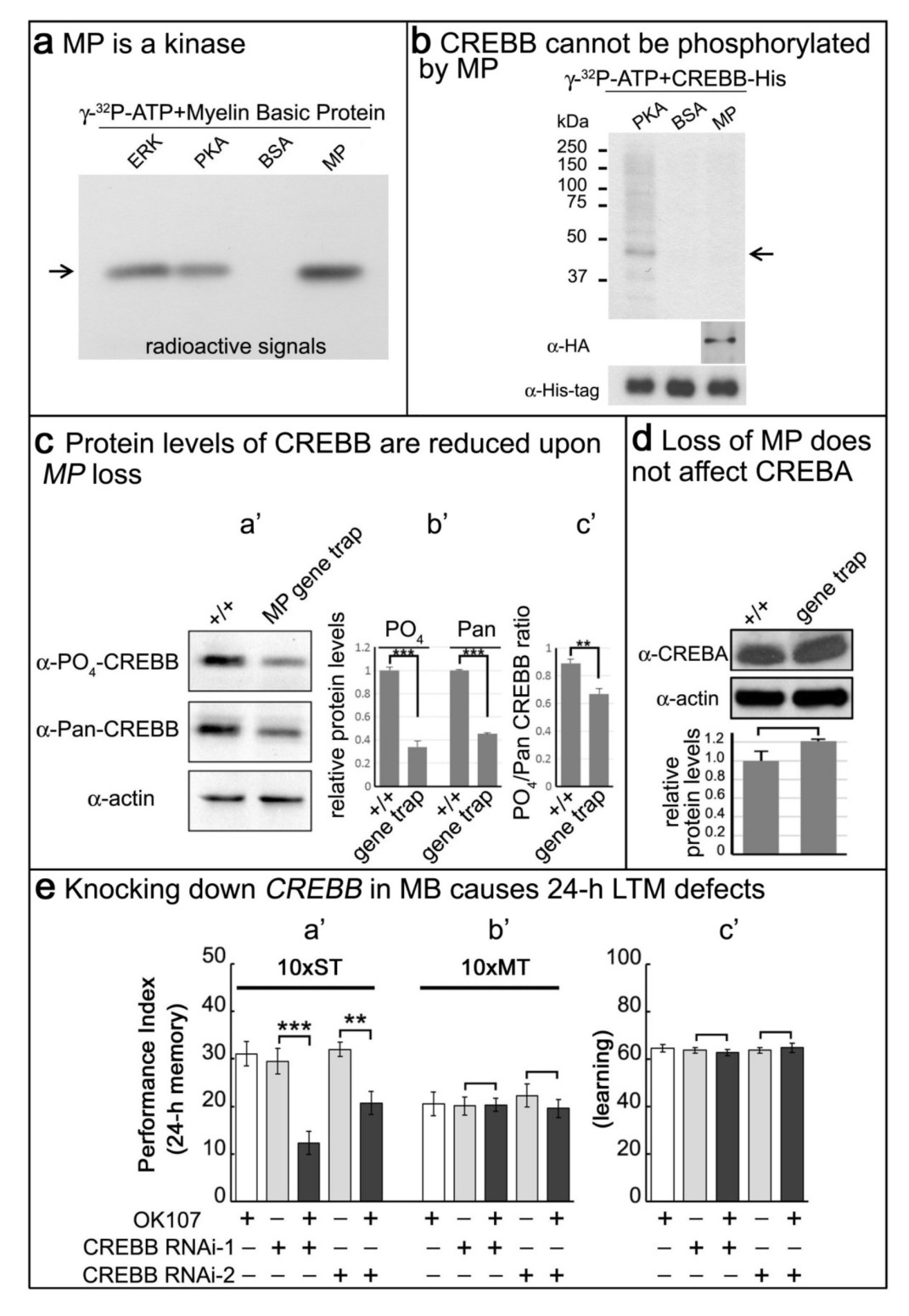

**Figure 4.** MP is a kinase, and loss of *MP* affects CREBB protein levels in *MP* gene trap animals. (**a**) MP can phosphorylate myelin basic protein similar to ERK and PKA. Arrow points to MBP. (**b**) CREBB is not phosphorylated by MP kinase. As a positive control, we used PKA to phosphorylate the CREBB-His tag protein. MP failed to phosphorylate CREBB under these conditions. BSA was used as a negative control. Arrow points to CREBB. (**c**) In (a'–b'), protein levels of phospho-CREBB and total CREBB in the absence of MP are decreased, and

*Figure 4 continued on next page*

*Figure 4 continued*
also the phosphorylation ratio of phospho-CREBB/total CREBB in (c'). Total CREBB was detected by anti-Pan-CREBB, phosphorylated CREBB was detected by anti-PO$_4$-CREBB (**p<0.01. ***p<0.001). (d) Protein levels of CREBA are not affected in MP gene-trap flies. CREBA protein was detected by anti-CREBA. The protein levels were normalized to actin. (Student's *t* test, n = 3). (e) Knocking down CREBB in MB with CREBB RNAi causes 24 hr LTM defects (10x ST, a'), but not ARM (10x MT, b') and learning in (c'). 10x ST = 10 times spaced training. 10x MT = 10 times massed training. The mean ±SEM is plotted for each genotype; n = 8 for each group. **p<0.01. ***p<0.001.
DOI: https://doi.org/10.7554/eLife.33007.012
The following figure supplement is available for figure 4:

**Figure supplement 1.** *CREBB* mRNA level is not affected in *MP* gene trap fly, but CREBB proteins are reduced.
DOI: https://doi.org/10.7554/eLife.33007.013

*MI03008* mutant brains. Hence, these data suggest that MP promotes the translation or stabilization of CREBB protein.

If *CREBB* mediates the effects of *MP* on LTM and loss of MP function reduces CREBB protein levels, then reducing CREBB in MB should phenocopy the effects of eliminating MP in MBs. To determine whether this is the case, we knocked down *CREBB* mRNA in adult MB using *OK107-GAL4* and two independent *UAS-RNAi* lines by shifting 3–5 day old flies kept at 18°C to 25°C three days prior to testing. We find that decreasing CREBB in MB causes a reduction of 24 hr LTM with both RNAi knockdowns (10 x ST; *Figure 4e,a'*), but it does not affect learning (*Figure 4e,c'*) or ARM (10 x MT; *Figure 4e,b'*). Comparison of these results with the knockdown of MP in MBs (*Figure 2*) indicates that loss of MP or CREBB function results in very similar memory defects. The most parsimonious explanation of these data is that loss of the MP protein kinase leads to reduced levels of CREBB protein, which in turn leads to a severe LTM impairment.

## MP is regulated by PKA

The MP protein contains the canonical PKA phosphorylation motif RRFS (*Huang et al., 2005*) at residues 331–334 (*Figure 5—figure supplement 1*). To determine whether serine residue 334 is a substrate of PKA we introduced a S334A mutation into MP (MP$^{S334A}$) and examined phosphorylation of both the wild-type and mutant proteins by PKA. As shown in *Figure 5a*, MP is phosphorylated by PKA, but MP$^{S334A}$ is barely phosphorylated. We conclude that S334 is required for proper MP phosphorylation by PKA. To determine whether phosphorylation alters the MP kinase activity, we next assayed the kinase activities of both wild-type MP and MP$^{S334A}$. As shown in *Figure 5b*, the kinase activity of MP$^{S334A}$ is significantly lower than that of wild-type MP.

The above data indicate that PKA activates MP in addition to activating CREBB. Loss of function mutations in *PKA* cause a subtle but significant reduction in CREBB (*Figure 5—figure supplement 2*). Hence, PKA and MP may work together in a coherent feedforward loop (*Mangan et al., 2003*) to upregulate CREBB activity and support LTM formation. To assess possible synergistic interactions between *PKA* and *MP*, we simultaneously reduced the protein levels of both MP and the catalytic subunit of PKA by creating *MI03008,Pka+/MP+,Pka-C1$^{B10}$* heteroallelic flies, and assessed the levels of CREBB protein by Western blot (*Fropf et al., 2013*). CREBB immunoreactivity in the heads of these animals is very severely reduced compared to that observed in the heads of either wild-type contols or animals with reduced gene dosage of only *MP* or *Pka-C1* (*Figure 5c*). Simultaneous reduction of *PKA* and *MP* thus substantially potentiates the effects of reducing either protein alone, consistent with a model in which the two kinases act within the same signaling pathway to regulate CREBB activity and LTM. To determine if the *MI03008,Pka+/MP+,Pka-C1$^{B10}$* animals exhibit memory defects, we assayed their ability to both learn and form 24 hr LTM. We find that these flies exhibit normal learning (*Figure 5d,a'*) and ARM (10 x MT; *Figure 5d,c'*), but lack 24 hr LTM (10 x ST; *Figure 5d,b'*) consistent with the loss of *CREBB* function. Interestingly, flies heterozygous for either *MP* or *Pka-C1* display neither learning nor LTM defect, suggesting a potent synergistic interaction and feedforward loop between *PKA* and *MP*.

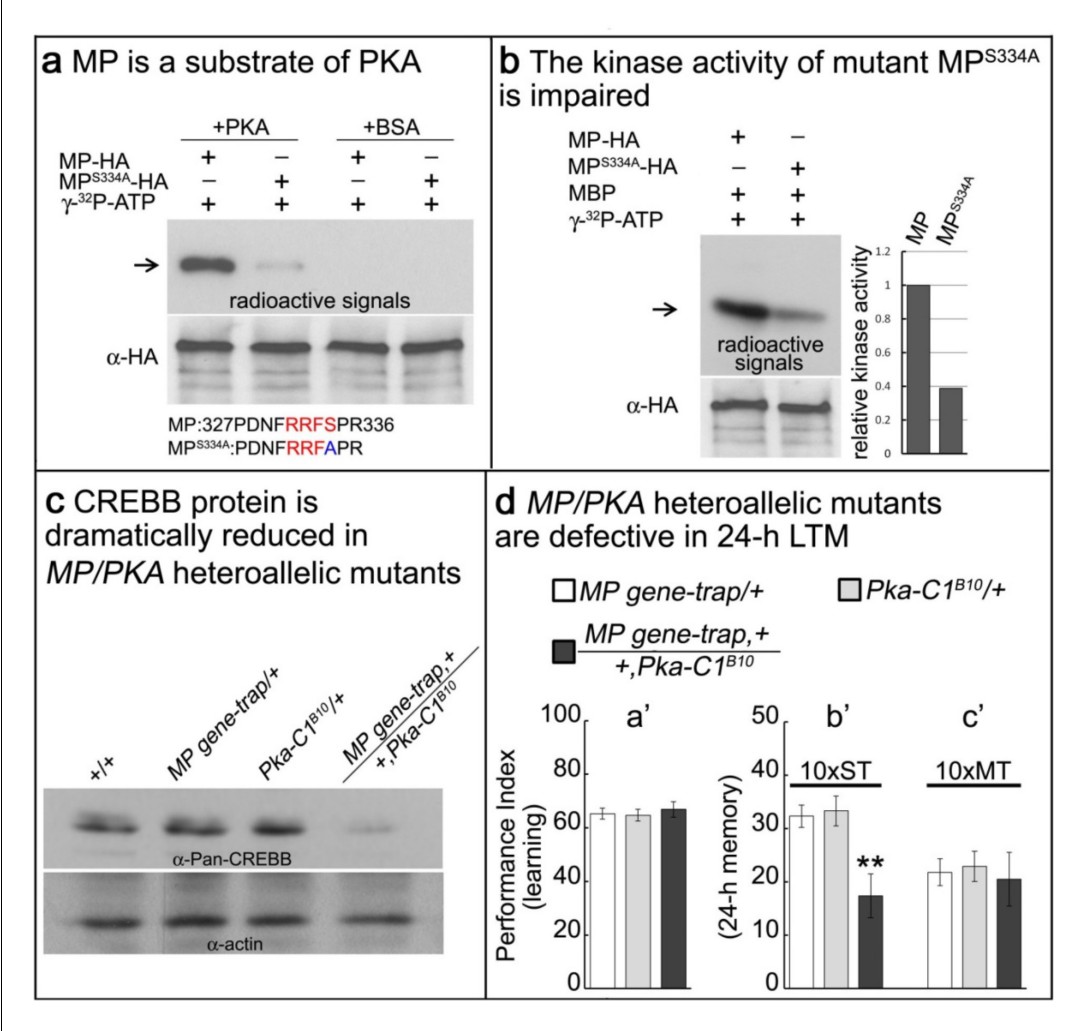

**Figure 5.** MP kinase is regulated by PKA. (a) The potential PKA phosphorylation site of MP was altered to S334A. MP can be phosphorylated by PKA, but MP$^{S334A}$ is severely impaired. Arrow points to wild-type and mutant MP. (b) MP$^{S334A}$ has severely reduced kinase activity compared to wild-type MP. Arrow points to MBP. (c) CREBB protein levels are dramatically reduced in *MP gene-trap +/+Pka-C1$^{B10}$* heteroallelic flies. CREBB protein levels are probed with anti-Pan-CREBB antibody. (d) *MP gene-trap +/+Pka-C1$^{B10}$* heteroallelic flies have intact learning in (a′) and ARM in (c′), but show 24 hr LTM impairment in (b′). The mean ±SEM is plotted for each genotype; n = 8 for each group. \*\*p<0.01.

DOI: https://doi.org/10.7554/eLife.33007.014

The following figure supplements are available for figure 5:

**Figure supplement 1.** Human SBK1 and fly MP (CG11221) both contain a PKA phosphorylation site.

DOI: https://doi.org/10.7554/eLife.33007.015

**Figure supplement 2.** Protein levels of CREBB are reduced in *Pka-C1* mutants.

DOI: https://doi.org/10.7554/eLife.33007.016

## Discussion

Using MiMIC technology, we converted 27 genes encoding putative protein kinases with the Trojan *T2A-GAL4* exon and performed an image screen for genes expressed in MBs (*Diao et al., 2015*). This tagging approach is especially useful for genes that are expressed at low levels in the CNS. By tagging the proteins with GFP, a conditional and reversible knockdown can be achieved in almost any tissue or cell (*Nagarkar-Jaiswal et al., 2015*). This allowed us to identify a novel serine/threonine protein kinase, *Meng-Po* (*MP*), that is a critical player in LTM formation in *Drosophila. MP* is a homologue of *SBK1* in mammals (*Figure 1—figure supplement 4*), a gene that is expressed in the hippocampus and the cortex (*Nara et al., 2001*; *Skarnes et al., 2011*). Loss of this gene in mice is

associated with embryonic lethality (*Skarnes et al., 2011*), whereas in flies, loss of *MP* leads to a reduction in viability as well as sterility.

Our data show that CREBB stability is highly susceptible to loss of MP. CREBB activity is modulated by phosphorylation via PKA and CamKII in *Drosophila* (*Horiuchi et al., 2004*). Although our findings indicate that MP kinase activity is critical for maintaining CREBB levels and that MP kinase activity acts in synergy with PKA (*Figure 6*), we have not been able to demonstrate that CREBB is a direct target of MP. However, some kinases require a previously phosphorylated residue as part of their recognition sequence and we have not mixed various kinases with MP in our in vitro assays (*Horiuchi et al., 2004*). Hence, it remains to be established how CREBB is degraded in the absence of MP.

A reduction in CREB levels has been shown to be associated with an age-dependent memory loss in rodents. Interestingly, delivery of CREB protein in the hippocampus using somatic cell transfer attenuated LTM impairement (*Mouravlev et al., 2006*). However, no gene has so far been shown to affect CREBB stability in vivo and our findings that MP, together with PKA, synergize to dramatically affect CREBB levels via a feedforward loop (*Mangan et al., 2003*), reveal another mechanism to control CREBB levels during memory formation (*Figure 6*). This model is supported by the observation that overexpression of MP increases CREBB activity and promotes memory formation, suggesting that it is a central player in LTM.

## Materials and methods

### Fly strains

Fly strains were maintained on standard cornmeal-yeast-agar medium at 25°C, at 60–70% relative humidity and on a 12/12 hr light/dark cycle. The *MiMICs* were created in the Bellen lab (*Diao et al., 2015*; *Nagarkar-Jaiswal et al., 2015*; *Venken et al., 2011*) and *P247-GAL4* (*Zars, 2000*), *OK107-GAL4* (*Pascual and Préat, 2001*), *Tub-GAL80ts*, *Pka-C1H2*, *PKa-C1B10*, *UAS-FLP*, *20xUAS-6xmCherry*

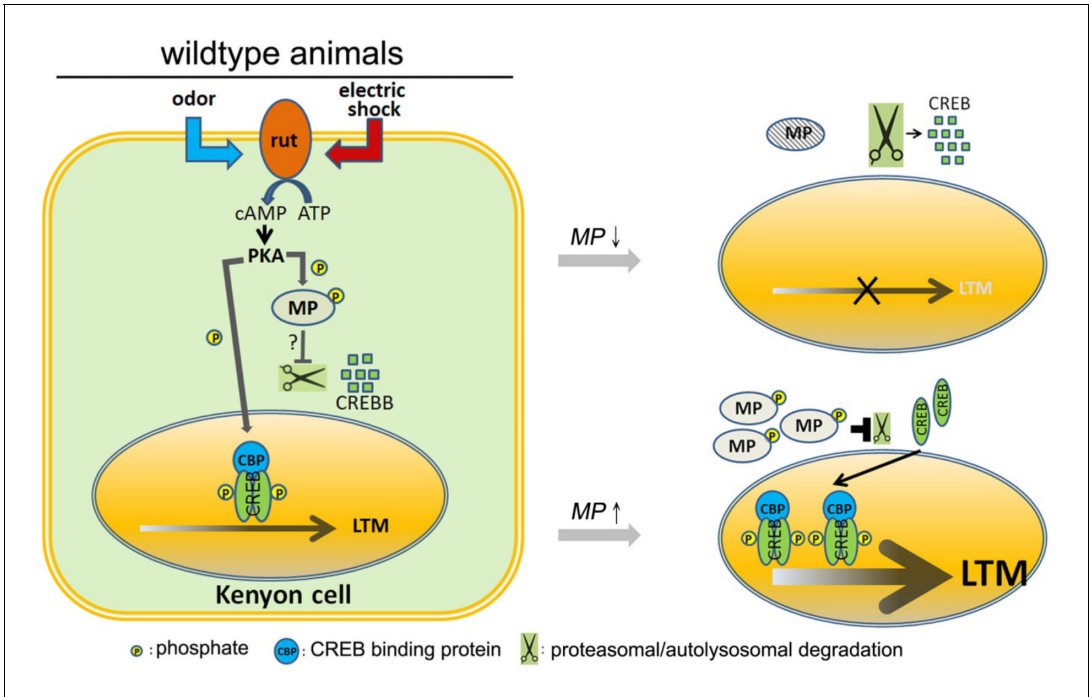

**Figure 6.** Model. In wild-type animals, *rutabaga* (*rut*) as a coincidence detector can receive the odor and electric shock signals then activate cAMP-PKA signaling. MP is phosphorylated by PKA which maintains the CREBB levels, possibly by inhibiting proteasomal or autolysosomal degradation. It permits CREB-dependent LTM to form. Knocking down *MP* will unlock this inhibition and facilitate CREBB degradation thereby disrupting LTM. In contrast, overexpression of MP promotes LTM formation.

DOI: https://doi.org/10.7554/eLife.33007.017

*and UAS-mCD8::GFP* (*Pfeiffer et al., 2010*) were obtained from the Bloomington Drosophila Stock Center (USA). The *Canton-S w^1118^* (iso1CJ) wild-type fly (*Lee et al., 2011*) was from Josh Dubnau. The *CRE-F-Luc* fly was from Jerry Yin (*Tanenhaus et al., 2012*). *UAS-RNAi* flies were from BDSC or VDRC. The stock information is listed on *Supplementary file 1*.

## Plasmid constructs

The MP-HA and CREBB-6xHis cDNA were synthesized by GenScript. The cDNA fragments were double digested with *EcoR*I and *Xho*I and cloned into pUAST-attB. For site-directed mutagenesis of *MP*, the primer sets below were used:

MP$^{S334A}$-F: 5'-CGCCGCTTCGCCCCCCGCCTGAT-3'
MP-R: 5'-CTTCAACAGCCCGTGGAACCACC-3'

The PCR reaction was performed with the Q5 Site-Directed Mutagenesis kit (NEB). The PCR product was ligated and transformed into *E.coli* competent cells, and the colonies were selected on ampicillin/LB agar plates. For transgenic animals, the *UAS-MP-HA* plasmid was extracted with HiPure Plasmid Midiprep kit (Invitrogen) for microinjection.

## RT-PCR

Total RNA of flies was extracted with the RNAspin Mini kit (GE Healthcare). RT-PCR was performed with OneStep RT-PCR kit (QIAGEN). For RT-PCR, the primer sets below were used:

MP-RT-F: 5'-GAAAACAAGTCTTCAGAAATGGGCACTATCG-3'
MP-RT-R: 5'-AAAGTCCGGCGTGAAGACCAGGATAT-3'
CREBB-RT-F: 5'-ACAACAGCATCGTCGAGGAGAACG-3'
CREBB-RT-R: 5'-CGTGTTCGGTTCGGGCTTGATCTT-3'
rp49-RT-F: 5'-CCAAGGACTTCATCCGCCACC-3'
rp49-RT-R: 5'-GCGGGTGCGCTTGTTCGATCC-3'

## Confocal imaging

Image processing was performed as described previously (*Lee et al., 2011*). Briefly, dissected brains were fixed in PBS with 4% paraformaldehyde at 4°C overnight, transferred to PBS with 2% Triton X-100 at room temperature, vacuumed for 1 hr and left overnight in the same solution at 4°C. For immunostaining of GFP, the samples were incubated with anti-GFP antibody conjugated with FITC (1:500) (Abcam) in PBS with 0.5% Triton X-100 overnight. Brains were cleared and mounted in Rapi-Clear (SunJin Lab Co.) and imaged with a Zeiss LSM 880 Confocal Microscope under a 20 x or 40 x C-Apochromat water immersion objective lens.

## Overexpression and pulldown of MP, MP$^{S334A}$ and CREBB in S2 cells

The Effectene transfection reagent (QIAGEN) was used to deliver DNA to S2 cells. For protein over-expression, $2 \times 10^6$ S2 cells were transferred into 5 ml fresh media overnight (Schneider's Drosophila medium, Gibco). The cells were collected by centrifuging them for 2 min. *UAS-MP-HA*, *UAS-MP$^{S334A}$-HA* or *UAS-CREBB-6xHis* were co-transfected with *Act-GAL4* into S2 cells. Transfected cells were placed at room temperature for two days in medium, collected, and lysed with sample lysis buffer (50 mM Tris-Cl pH 7.5, 125 mM NaCl, 5% glycerol, 1% NP40, 1.5 mM MgCl$_2$, 0.2 mM DTT) containing a protease inhibitor mix (cOmplete, Roche). The cell lysate was collected and α-HA agarose (EZview Red Anti-HA Affinity Gel, Sigma) or α-His resin (HisPur Ni-NTA Resin, ThermoFisher) was added for protein pulldown.

## Western blot

Fly heads were collected and mashed in a sample lysis buffer. The samples with SDS sample buffer were boiled and run on a 10% SDS-PAGE gel and transferred to nitrocellulose membranes. Primary antibodies for PO$_4$-CREBB (1:1000), pan-CREBB (1:5000) (*Fropf et al., 2013*), CREBA (1:5000) (*Andrew et al., 1997*) (Developmental Studies Hybridoma Bank), HA-tag (1:2000) (BioLegend), His-tag (1:2000) (Clontech) and actin (1:10,000) (Abcam) were used and detected by HRP conjugated secondary antibody (1:10,000) (Jackson ImmunoResearch). For detecting the ATG2-CREBB protein (*Fropf et al., 2013*), 50 adult brains were dissected and collected in 20 μL of the sample lysis buffer and placed at −80°C for overnight. After vortexing for one minute, samples were mixed with SDS

sample buffer and boiled before loading into an SDS-PAGE gel and transferred to nitrocellulose. Primary antibodies for ATG2-CREBB (1:1,000) (*Fropf et al., 2013*) was used and detected by HRP conjugated secondary antibody (1:10,000) (Jackson ImmunoResearch). An ECL reagent kit (ThermoFisher) was used for Western blot detection.

### Kinase assay

ERK and PKA were purchased from NEB. MP and MP$^{S334A}$ were overexpressed and purified from S2 cells. For kinase assay, the kinases (ERK: 0.01U (NEB), PKA: 0.1U (NEB), MP and mutant MP: 13.5 µl of α-HA-agarose pulldown) were added into a cocktail containing the kinase buffer (0.5 mg/ml BSA, 15 µM Tris-Cl pH7.5 in final) $MgCl_2$/ATP (0.2 mM in final), $\gamma$-$^{32}$P-ATP (10 µCi in final), and myelin basic protein (MBP) (1.2 mM in final). For testing if MP or CREBB are substrates of PKA, MP and CREBB were pulled down, and incubated at 30°C for 30 min. The reactions were terminated by adding 2x SDS sample buffer, boiling for 5 min, and run on SDS gels. After transferring to nitrocellulose membranes, radioactive signals were detected by CL-X posure film (ThermoFisher).

### Luciferase activity assay

The luciferase assay system was from Promega. All groups of flies were raised in the same incubator in 12 hr light/12 hr dark conditions, and the CRE-luciferase assay was performed at the same time for all genotypes tested to avoid a circadian effect (*Fropf et al., 2014*). Briefly, fly heads were collected and homogenized in a reporter lysis buffer (10 heads/20 µl) and placed at −80°C overnight. The supernatant of lysate was collected and mixed with luciferase assay reagents (20 µl lysate/100 µl luciferase assay reagent) at RT for 1 min in 96-well plate. The luciferase activity was analyzed using an *Optima* luminescence reader (BMG LABTECH).

### Behavioral assay

Behavioral assays were performed with balanced comparative groups which were trained and tested in parallel without blinding. For behavior assays, *MB-GAL4* flies were outcrossed with *Canton-S w$^{1118}$* (iso1CJ) or *y w* flies for at least five generations, and *Tub-Gal80$^{ts}$*, *UAS-RNAi* and *UAS-MP-HA* flies were outcrossed with iso1CJ. For RNAi knockdown, flies were raised at 18°C until eclosion then transferred to 25°C for 3 days. Aversive olfactory learning was performed using the T-maze apparatus (*Tully and Quinn, 1985*). In brief, one training session consists of approximately 100 flies that were electrically shocked while exposed to one of two odors (3-octanol and 4-methylcyclohexanol, Sigma). The shock was alternated between 3-octanol and 4-methylcyclohexanol. Flies trained by one training session and tested immediately are tested for learning, whereas those tested 3 hr later are assessed for 3 hr memory. Flies undergoing ten training sessions and tested 24 hr later are assessed for 24 hr memory or LTM. For cycloheximide (CXM, Sigma) feeding (*DeZazzo and Tully, 1995*), 35 mM CXM in 5% glucose was added on Whatman 3 MM filter paper in a bottle with fly food to feed flies overnight prior to performing training sessions. After training, the flies are placed in a vial containing CXM filter paper with fly food for another 24 hr prior testing.

### Statistics

Statistical analyses were performed by KaleidaGraph 4.1 (Synergy software). Behavioral data were evaluated via one-way ANOVA followed with Tukey's test for multiple comparisons (*Lee et al., 2011*). Data from two groups are analyzed by *t*-test. All data are presented as 'mean ±SEM'. *$p < 0.05$, **$p < 0.01$, ***$p < 0.001$.

## Acknowledgements

We thank Amir Fayyazuddin for providing very insightful suggestions. We thank Josh Dubnau, Jerry Yin, and the Bloomington *Drosophila* Stock Center (NIH P40OD018537) and Vienna *Drosophila* Resource Center for flies. We thank Jerry Yin for sharing the α-ATG2-, α-Pan- and α-PO$_4$-CREBB antibodies and the Developmental Studies Hybridoma Bank for α-CREBA antibodies. Confocal microscopy was supported by the BCM IDDRC Neurovisualization Core, which is supported by the NICHD (U54HD083092). We thank Jiangxing Lv, Yuchun He, Hongling Pan, Ying Fang, Qiaohong Gao, Zhihua Wang and Lily Wang for generating MiMIC *GFP-tag* and *T2A-GAL4* fly stocks. This

research was supported by NIH/NIGMS R01GM067858, the Robert A and Renee E Belfer Family Foundation and the Huffington Foundation. Hugo J Bellen is a Howard Hughes Medical Institute Investigator.

## Additional information

### Competing interests

Hugo J Bellen: Reviewing editor, *eLife*. The other authors declare that no competing interests exist.

### Funding

| Funder | Grant reference number | Author |
| --- | --- | --- |
| National Institute of General Medical Sciences | R01GM067858 | Pei-Tseng Lee |
| Eunice Kennedy Shriver National Institute of Child Health and Human Development | U54HD083092 | Hugo J Bellen |
| Robert A. and Renee E. Belfer Family Foundation | | Hugo J Bellen |
| Huffington Foundation | | Hugo J Bellen |
| Howard Hughes Medical Institute | | Hugo J. Bellen |

The funders had no role in study design, data collection and interpretation, or the decision to submit the work for publication.

### Author contributions

Pei-Tseng Lee, Conceptualization, Resources, Data curation, Validation, Methodology, Writing—original draft, Writing—review and editing; Guang Lin, Conceptualization, Data curation, Formal analysis, Validation, Methodology; Wen-Wen Lin, Fengqiu Diao, Resources; Benjamin H White, Resources, Writing—original draft, Writing—review and editing; Hugo J Bellen, Conceptualization, Resources, Data curation, Supervision, Funding acquisition, Validation, Investigation, Writing—original draft, Project administration, Writing—review and editing

### Author ORCIDs

Pei-Tseng Lee http://orcid.org/0000-0002-7501-7881
Guang Lin https://orcid.org/0000-0001-5594-3397
Benjamin H White https://orcid.org/0000-0003-0612-8075
Hugo J Bellen http://orcid.org/0000-0001-5992-5989

### Decision letter and Author response

Decision letter https://doi.org/10.7554/eLife.33007.021
Author response https://doi.org/10.7554/eLife.33007.022

## Additional files

### Supplementary files

• Supplementary file 1. Fly stocks and antibodies information.
DOI: https://doi.org/10.7554/eLife.33007.018
• Transparent reporting form
DOI: https://doi.org/10.7554/eLife.33007.019

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
