## [Decision Letter]

Thank you for sending your article entitled "A kinase-dependent feedforward loop affects CREBB stability and long-term memory formation" for peer review at *eLife*. Your article has been favorably evaluated by K VijayRaghavan as the Senior and Reviewing Editor and two peer reviewers.

Please can you, in particular, see the revisions requested by reviewer #2, which require additional experiments so let us know if they are addressable and if so, your timelines for doing so.

*Reviewer #1:*

This work identifies a novel gene, *Meng-Po (CG11221* in *Drosophila; SBK1* in human), involved in LTM formation through regulation of CREBB. This word is significant not only for identifying this novel gene but also for use of MIMIC technology and application of deGradFP method. The manuscript is suitable to publish in *eLife*, provided that concerns raised below are sufficiently addressed.

1) Overview of the proteins required for olfactory aversive learning/memory formation in *Drosophila* is shown in Figure 1—figure supplement 1. However, the information included is incomplete and, in some aspects, is inaccurate. Therefore, this figure is better to be removed.

2) To conclude that MP does not affect learning, the authors should provide experiments with weak training intensity using reduced voltage or pulses of electric shock so as to rule out the potential ceiling effects.

3) As ASM and LTM are different forms of memory, although the authors provide evidence in supporting of restoration of 3-h memory defect through reintroduction of MP. It would be necessary to confirm whether LTM defect can be restored by the same manipulation, for the focus of this manuscript is on LTM.

4) To validate LTM is either reduced (Figure 2) or enhanced (Figure 3) by manipulations of MP, the standard approach in the field is to confirm whether the affected memory components are protein-synthesis dependent. Such experiments should be included.

*Reviewer #2:*

This manuscript describes the identification and functional testing of a new, but conserved, protein kinase in *Drosophila* memory formation. Although heavily studied in neuronal function, *new* "upstream" information that affects CREB activity is rare, and this work adds to this small fraction of the more than 12,000 PubMed citations on CREB. The authors demonstrate that MP activity is required for long-term memory (LTM) formation, but not learning or anesthesia-resistant memory (ARM). It's requirement in LTM formation is most likely because of an indirect effect on CREBB (or dCREB-2) stability or translation. Protein kinase A phosphorylates MP in vitro, and they probably work together to affect the levels of CREBB protein. The experiments are well conceived, although there is some confusion about the design of the behavioral experiments that were done. This issue is a major one *if* the different groups that constitute a figure were not done simultaneously. One behavioral control experiment is missing (regardless of whether the different groups were all done simultaneously or not).

The other suggested experiment would clarify that removing the MP kinase decreases the CREBB (or dCREB2) transcriptional activity, or the amount of the activator subspecies, consistent with its effect on memory formation. One version of this experiment is straightforward and requires the use of published antibodies. The alternative reporter-based experiment is also straightforward but requires more time to complete due to the genetics that needs to be done. (comments #1-2)

The rest of the comments just require changes to verbiage or adding some background verbiage, and some simple answers to experimental details that are missing. (comments #3-6)

1)The presentation of data is confusing. For Figure 2, Figure 3, Figure 4 and Figure 5, it is very hard to tell if all of the different experimental groups were trained and tested together, or separately in subgroups, as implied by the spacing between the sets of histograms. For example, in Figure 2 (middle set of histograms) there are 9 different genotypes/induction regimens/treatments that are shown. Were all 9 groups done together, or have the authors grouped them to make statistical comparisons easier? In Figure 3, Panel c', there are 6 different experimental groups, while in 3B, Panel d' there are 10 different groups. If they were done all at once, this should be stated, and the comparisons should be presented so that the reader can walk through the results one parameter at a time. For example, for Figure 2 (middle panel), numbering the histograms 1-9 (as well as having the color code) and referencing the comparisons in the text would be helpful. The significance could then be described in the figure legend (e.g. 2 vs. 3, ** p < 0.01; 3 vs. 7, 8 or 9 etc.). This would facilitate the comparison between groups if they are referenced (by number) in the text.

More traditionally, when the experiments are done with a subset of the samples (one parameter at a time), this reviewer counts 4 different experiments presented in Figure 2 (middle set of histograms): a) the normal score for 3h memory when flies are maintained at 18 degrees, b) the effect of genotype on 3h memory when flies are shifted to 28 degrees, c) the effect of temperature shift on ARM/ASM, and d) the recovery when flies are shifted from 28 back down to 18 degrees. For experiment (a), a control that is missing is the demonstration that the wild-type and experimental flies have a normal distribution of ARM/ASM (+ cold shock) when they are maintained at 18 degrees. Experiment (b) would involve groups 3, 7, 8, and 9. Experiment (c) would involve groups 3, 4, 5, and 7, while experiment (d) would involve groups 2, 5 and 6. However, each separate experiment should have the appropriate controls.

Whether done simultaneously or in subgroups, the control for experiment (a) [+/- cold shock on the wild-type background and experimental flies at 18 degrees] needs to be done. This experiment would show that there is nothing in the background stock that disrupts the normal fractional distribution of ARM and ASM.

2) To complete the demonstration that mutant MP decreases CREBB (or dCREB2) activity, the authors need to either show that CRE-luciferase activity decreases in the MP mutant background, or that the amount of the dCREB2 nuclear activator decreases in the mutant background. The first can be done using a mushroom body-driven FLP transgene, combined with the reporter. This binary combination would allow the authors to test the MP mutant (or wild-type chromosome) for an effect on luciferase activity that only originates from the mushroom body.

For the second possibility, a simple western blot could suffice. Numerous molecular species are made off of the CREBB (or dCREB2) gene, and it is clear that the MP mutant decreases the immunoreactive species shown in their western blots. However, since the relative mobility of molecular weight markers is not shown on the western blots, it is hard to determine if the effects are on the more abundant (and larger) blocker isoforms, the less abundant and faster migrating activator isoforms, or both. It is most likely that the authors show decreases in the blocker (35-40 kd) species. However, in order for their behavioral results to make sense, there should be a decrease in both activator and blocker species in the MP mutant background. The authors should be able to detect a decrease in the amount of the nuclear activator species in their mutant background. This requires using antibodies reported in Fropf et al. (2013, 2014) and making and assaying crude nuclear extracts. This experiment will need considerably more starting material, but that should be easy to obtain for the wild-type and mutant backgrounds. Demonstrating that both blocker and activator decrease in the MP mutant background would complete their correlation/causation links between MP and CREBB.

---

## [Author Response]

Reviewer #1:[…] 1) Overview of the proteins required for olfactory aversive learning/memory formation in Drosophila is shown in Figure 1—figure supplement 1. However, the information included is incomplete and, in some aspects, is inaccurate. Therefore, this figure is better to be removed.

To accommodate the reviewer’s request we reorganized Figure 1—figure supplement 1. We added *msk, CASK* and *aPKC*. We now use *msk* to replace the *JNK*/MAPK pathway. Also, we added *CASK* as it has been reported to regulate *CamKII* activity. Given that *aPKC* has been reported to enhance 24-memory, but is specific for 24-ARM, we altered the figure and changed the figure legend to accommodate the changes. “These include *CamKII* (Akalal et al., 2010), *CASK* (Gillespie and Hodge, 2013), *msk* (MAPK pathway) (Li et al., 2016; Philip et al., 2001; Skoulakis and Davis, 1996), *ignorant (ign, S6kII*) (Putz et al., 2004), *wallenda (wnd*) (Huang et al., 2012) and *S6K*

(Fropf et al., 2013). *aPKC* has been shown to enhance 24-h memory in *Drosophila*. However, it enhances 24-h ARM, but not 24-h LTM (Drier et al., 2002).”. (Figure 1—figure supplement 1 and legend).

2) To conclude that MP does not affect learning, the authors should provide experiments with weak training intensity using reduced voltage or pulses of electric shock so as to rule out the potential ceiling effects.

To address this question, we knocked MP down in MB with RNAi and performed learning assays with a reduced number of electric shock pulses (6x instead of 12x). The results show that the learning performance is not different between controls and MP knockdown flies (Figure 1—figure supplement 3, learning, 30” CS+US). We edited the legend: “Reducing the number of training sessions from 12x to 6x electric shocks (from 60” to 30”) reveals that learning is still not affected in the absence of MP.” (Figure 1—figure supplement 3)

In the main text we added “To provide additional evidence that *CG11221* is not required for learning, we used a short CS/US association protocol (30” CS+US instead of 60”) to train flies. Knockdown of MP did not affect learning, further indicating that *MP* is not required for learning (Figure 1—figure supplement 3).”

3) As ASM and LTM are different forms of memory, although the authors provide evidence in supporting of restoration of 3-h memory defect through reintroduction of MP. It would be necessary to confirm whether LTM defect can be restored by the same manipulation, for the focus of this manuscript is on LTM.

To address the reviewer’s concern, we treated *MP-GFP; UAS-deGradFP/P247-GAL4* flies in which the MP protein was restored (c’ condition in Figure 2) and performed a 10x ST assay to assess 24-h LTM. The data are now included in Figure 2 at the bottom. These flies exhibit normal 24-h memory. We added: “This indicates that the loss of memory is protein synthesis-dependent. Loss of 24-h LTM memory can be restored when returning the flies to 18°C for two days after knockdown. (Figure 2 memory, 10x ST, c’).”.

4) To validate LTM is either reduced (Figure 2) or enhanced (Figure 3) by manipulations of MP, the standard approach in the field is to confirm whether the affected memory components are protein-synthesis dependent. Such experiments should be included.

To address the reviewer’s concern, we used deGradFP to knock down MP-GFP in MB (b’ condition) and fed the flies cycloheximide 24hrs prior and after training to impair new protein synthesis. As shown in Figure 2 (10xST, b’+CXM, red), MP-GFP knockdown flies performed similarly in a 24-h memory test when compared to control flies. The data indicate that MP is required for protein synthesis dependent memory. We added the following sentence: “After treatment with 35 mM cycloheximide (CXM), an inhibitor for protein synthesis, the MP knockdown flies (Figure 2 memory, 10x ST, b’+CXM, red column) do not perform worse than other control flies. This indicates that the loss of memory is protein synthesis-dependent.”

To address if the memory enhancement observed when we overexpress MP is dependent on protein synthesis we performed the following experiments. We overexpressed MP in the MB neurons at 30°C for three days. We then fed these flies cycloheximide for 24 hrs prior and after training and tested LTM 24hrs later using the 3x ST paradigm and found that MP overexpression did not improve 24-h memory (Figure 3’, 3xST, b’+CXM, red). We added the following sentence: “The observed memory enhancement can be erased by feeding flies 35 mM CXM, indicating that enhanced memory is protein synthesis-dependent (3x ST+CXM;Figure 3’).” In the legend of Figure 3, we added: “However, memory enhancement can be erased by feeding flies with cycloheximide.” We added sentences in Materials and methods: “For cycloheximide (CXM, Σ) feeding (DeZazzo and Tully, 1995), 35 mM CXM in 5% glucose was added on Whatman 3 MM filter paper in a bottle with fly food to feed flies overnight prior to performing training sessions. After training, the flies are placed in a vial containing CXM filter paper with fly food for another 24 hrs prior testing.”.

Reviewer #2:This manuscript describes the identification and functional testing of a new, but conserved, protein kinase in Drosophila memory formation. Although heavily studied in neuronal function, new "upstream" information that affects CREB activity is rare, and this work adds to this small fraction of the more than 12,000 PubMed citations on CREB. The authors demonstrate that MP activity is required for long-term memory (LTM) formation, but not learning or anesthesia-resistant memory (ARM). It's requirement in LTM formation is most likely because of an indirect effect on CREBB (or dCREB-2) stability or translation. Protein kinase A phosphorylates MP in vitro, and they probably work together to affect the levels of CREBB protein. The experiments are well conceived, although there is some confusion about the design of the behavioral experiments that were done. This issue is a major one if the different groups that constitute a figure were not done simultaneously. One behavioral control experiment is missing (regardless of whether the different groups were all done simultaneously or not).The other suggested experiment would clarify that removing the MP kinase decreases the CREBB (or dCREB2) transcriptional activity, or the amount of the activator subspecies, consistent with its effect on memory formation. One version of this experiment is straightforward and requires the use of published antibodies. The alternative reporter-based experiment is also straightforward but requires more time to complete due to the genetics that needs to be done. (comments #1-2)The rest of the comments just require changes to verbiage or adding some background verbiage, and some simple answers to experimental details that are missing. (comments #3-6)1)The presentation of data is confusing. For Figure 2, Figure 3, Figure 4 and Figure 5, it is very hard to tell if all of the different experimental groups were trained and tested together, or separately in subgroups, as implied by the spacing between the sets of histograms. For example, in Figure 2 (middle set of histograms) there are 9 different genotypes/induction regimens/treatments that are shown. Were all 9 groups done together, or have the authors grouped them to make statistical comparisons easier? In Figure 3, Panel c', there are 6 different experimental groups, while in 3B, Panel d' there are 10 different groups. If they were done all at once, this should be stated, and the comparisons should be presented so that the reader can walk through the results one parameter at a time.

We trained and tested all the grouped genotypes (brackets on top) at the same time and the same day. We now rearranged the data in Figure 2 and Figure 3 to show which experiments were performed in groups. The data in Figure 4 and Figure 5 are obvious.

For example, for Figure 2 (middle panel), numbering the histograms 1-9 (as well as having the color code) and referencing the comparisons in the text would be helpful. The significance could then be described in the figure legend (e.g. 2 vs. 3, ** p < 0.01; 3 vs. 7, 8 or 9 etc.). This would facilitate the comparison between groups if they are referenced (by number) in the text.

We tried this suggestion, but it created a very complicated figure. We kept the original format and rearranged the data in Figure 2 to make it clear which flies were tested at the same time.

More traditionally, when the experiments are done with a subset of the samples (one parameter at a time), this reviewer counts 4 different experiments presented in Figure 2 (middle set of histograms): a) the normal score for 3h memory when flies are maintained at 18 degrees, b) the effect of genotype on 3h memory when flies are shifted to 28 degrees, c) the effect of temperature shift on ARM/ASM, and d) the recovery when flies are shifted from 28 back down to 18 degrees.

The reviewer is correct and this is indeed precisely what we did.

For experiment (a), a control that is missing is the demonstration that the wild-type and experimental flies have a normal distribution of ARM/ASM (+ cold shock) when they are maintained at 18 degrees.

To address the reviewer’s criticism we tested wild-type and experimental flies in Figure 2 using a cold shock (in a’). We also added a 3-h memory/cold shock assay in the b’ condition for wildtype flies as well as *MP-GFP; UAS-deGradFP/P247-GAL4* and other control flies (these genotypes are marked with a black dot below). The results are shown in Figure 2 (Figure 2, Figure 3 memory+cold-shock).

Experiment (b) would involve groups 3, 7, 8, and 9. Experiment (c) would involve groups 3, 4, 5, and 7, while experiment (d) would involve groups 2, 5 and 6. However, each separate experiment should have the appropriate controls. Whether done simultaneously or in subgroups.

At the reviewer’s suggestion, each experiment was done with the appropriate controls and the data are now included in Figure 2. However, we would like to state that we originally performed the behavioral assay with the same flies using deGradFP to show protein knockdown and reversibility, and that should have obviated these control experiments. Given that it is the first time that we did perform the experiments with the deGradFP system we performed all the control experiments as requested by the reviewer. The results are exactly as anticipated.

The control for experiment (a) [+/- cold shock on the wild-type background and experimental flies at 18 degrees] needs to be done. This experiment would show that there is nothing in the background stock that disrupts the normal fractional distribution of ARM and ASM.

We performed this control and it is included in Figure 2. We also added a learning assay at 18 °C (Figure 2, learning, a’ condition).

2) To complete the demonstration that mutant MP decreases CREBB (or dCREB2) activity, the authors need to either show that CRE-luciferase activity decreases in the MP mutant background, or that the amount of the dCREB2 nuclear activator decreases in the mutant background. The first can be done using a mushroom body-driven FLP transgene, combined with the reporter. This binary combination would allow the authors to test the MP mutant (or wild-type chromosome) for an effect on luciferase activity that only originates from the mushroom body.For the second possibility, a simple western blot could suffice. Numerous molecular species are made off of the CREBB (or dCREB2) gene, and it is clear that the MP mutant decreases the immunoreactive species shown in their western blots. However, since the relative mobility of molecular weight markers is not shown on the western blots, it is hard to determine if the effects are on the more abundant (and larger) blocker isoforms, the less abundant and faster migrating activator isoforms, or both. It is most likely that the authors show decreases in the blocker (35-40 kd) species. However, in order for their behavioral results to make sense, there should be a decrease in both activator and blocker species in the MP mutant background. The authors should be able to detect a decrease in the amount of the nuclear activator species in their mutant background. This requires using antibodies reported in Fropf et al. (2013, 2014) and making and assaying crude nuclear extracts. This experiment will need considerably more starting material, but that should be easy to obtain for the wild-type and mutant backgrounds. Demonstrating that both blocker and activator decrease in the MP mutant background would complete their correlation/causation links between MP and CREBB.

We thank the reviewer for this suggestion. We tested the level of ATG2-CREBB in adult brains of wild type and MP mutants and also tested the levels with Pan-CREBB. We were able to detect the potential CREBB activator with the ATG2-CREBB antibody published by Fropf et al. (2013). The results are shown in Figure 4—figure supplement 1. We find that both the Pan-CREBB and ATG2-CREBB anti-bodies detect less CREBBs when MP is reduced as suggested by the reviewer and consistent with our previous data. We added: “We also tested if ATG2-CREBB is affected. […] As shown in Figure 4—figure supplement 1, we find that both the anti-Pan-CREBB antibody and the antibody recognizing ATG2-CREBB (Fropf et al., 2013) identify a ~30 kDa band that is reduced in *MI03008* mutant brains.’.

We added sentences in Materials and methods: ‘For detecting the ATG2-CREBB protein (Fropf et al., 2013), 50 adult brains were dissected and collected in 20 μL of the sample lysis buffer and placed at -80 °C for overnight. […] Primary antibodies for ATG2-CREBB (1:1,000)

(Fropf et al., 2013) was used and detected by HRP conjugated secondary antibody (1:10,000) (Jackson ImmunoResearch).’. In Figure 4—figure supplement 1, we added: “(d) Protein levels of Pan- and ATG2-CREBBs are reduced in *MP* mutant flies. Pan- and ATG2-CREBB protein levels were assessed with anti-Pan- and anti-ATG2-CREBB antibodies. Actin was used as an internal control.”. As recommended by the reviewer we also attempted to detect the proteins in nuclear fractions but failed to detect CREBB with Pan-CREBB and ATG2-CREBB antibodies.